# Hybrid Path Planning Combining Potential Field with Sigmoid Curve for Autonomous Driving

**DOI:** 10.3390/s20247197

**Published:** 2020-12-16

**Authors:** Bing Lu, Hongwen He, Huilong Yu, Hong Wang, Guofa Li, Man Shi, Dongpu Cao

**Affiliations:** 1National Engineering Laboratory for Electric Vehicles, Beijing Institute of Technology, Beijing 100081, China; 3120160195@bit.edu.cn (B.L.); 3120185272@bit.edu.cn (M.S.); 2Department of Mechanical and Mechatronics Engineering, Waterloo University, Waterloo, ON N2L 3G1, Canada; huilong.yu@uwaterloo.ca (H.Y.); dongpu.cao@uwaterloo.ca (D.C.); 3Tsinghua Intelligent Vehicle Design and Safety Research Institute, Tsinghua University, Beijing 100084, China; hong_wang@tsinghua.edu.cn; 4College of Mechatronics and Control Engineering, Shenzhen University, Shenzhen 518060, China; guofali@szu.edu.cn

**Keywords:** potential field, sigmoid curve, path planning, autonomous vehicles

## Abstract

The traditional potential field-based path planning is likely to generate unexpected path by strictly following the minimum potential field, especially in the driving scenarios with multiple obstacles closely distributed. A hybrid path planning is proposed to avoid the unsatisfying path generation and to improve the performance of autonomous driving by combining the potential field with the sigmoid curve. The repulsive and attractive potential fields are redesigned by considering the safety and the feasibility. Based on the objective of the shortest path generation, the optimized trajectory is obtained to improve the vehicle stability and driving safety by considering the constraints of collision avoidance and vehicle dynamics. The effectiveness is examined by simulations in multiobstacle dynamic and static scenarios. The simulation results indicate that the proposed method shows better performance on vehicle stability and ride comfortability than that of the traditional potential field-based method in all the examined scenarios during the autonomous driving.

## 1. Introduction

Path planning as an essential part of the autonomous driving has been widely researched in recent years. The path planning layer of autonomous vehicles (AVs) can be classified into the global and local path planners according to the planning horizon [1,2]. The global path planners are mainly focused on the navigation with the optimal economic, the least congestion and the highest average speed by considering the entire configurable space from the start point to the target point [3]. Different from the global path planners, local path planners usually pay more attention to the improvements on the driving safety and the vehicle stability in the process of dynamic obstacle avoidance by considering the constraints of kinematics and dynamics, during autonomous driving [4].

Many planning algorithms of AVs are inherited from wheeled-robotics principles because of their similarities in structure and control. In the wheeled robotics community, path planning methodologies can be classified into four groups, including graph search-based, sampling-based, interpolation-based and numerical optimization-based [5]. The idea behind graph search-based methods is to construct a configurable state-space based on graph theory and then use different search strategies (e.g., Voronoi diagrams [6], the Dijkstras algorithm [7], the A${}^{*}$ algorithm [8] or the State Lattice algorithm [9]) to generate a discrete route with grid or lattice occupancy. Being different from graph search-based methods, sampling-based methods can be further categorized into stochastic sampling and deterministic sampling depending on the sampling space used. Deterministic sampling-based methods [10] require less computation cost than the stochastic sampling-based methods [11], as they sample in a semistructured space instead of an entire configuration action-space or state-space. There are some common features between graph-based and sampling-based methods. For example, the paths generated from both methods are connected by a series of discrete waypoints [12], and these paths need further smoothing for practical application in AVs or wheeled-robotics. Since continuous curvature is a necessary requirement of drivable paths, interpolation-based path planning methods have been developed to generate smooth and continuous-curvature routes based on different curve models, such as spline curves model [13], clothoid curve model [14], etc.

The feasible solutions of satisfying the smooth and drivable constraints are usually not unique. Thus, numerical optimization-based path planning methods are developed to obtain the optimal route based on designing an objective function [15], e.g., the shortest-distance, the highest-efficiency, the shortest-time, etc. A potential field-based path planning method (PFBM), as a typical numerical optimization approach, was proposed by establishing the attractive potential field (PF) around a target point and the repulsive potential fields around obstacles to realize the obstacle avoidance of a robot in [16]. A composite PF is established with the constructed repulsive and attractive PFs, to automatically guide a robot to the destination by searching the gradient descent direction. Up to now, the PFBM has been applied both in structured [17,18] and unstructured [19] environments for AV path planning. Traditional PFBMs are usually based on a known target point and span the entire discrete space, which makes them reasonable and efficient for robotics control in indoor or simple environments. However, these requirements are difficult to accurately determine for autonomous driving in practical road environments. Furthermore, there is no qualitative assessment of the reasonability of the paths generated using PFBM, especially in driving scenarios with multiple obstacles. Besides, the planned paths are very sensitive to the configuration of the parameters [16]. For example, if the parameters of the PF functions are inappropriately configured or the obstacles are located with short distances, the route generated using PFBM is likely to fall into local minima, resulting in rough and unexpected routes.

To address the above mentioned problems, in this paper we propose a hybrid potential field sigmoid curve method (HPFSM) as shown in Figure 1, which aims to optimize the planned path of PFBM and to achieve an expected collision-free trajectory to improve the performance of autonomous driving. Firstly, the PF functions are designed based on the geometric shape, relative position and distance, etc. Then, the potential fields are established for the obstacles, the target lane and the road boundaries according to the defined PF functions, respectively. A collision-free route with the minimum PF can be achieved by fusing the established PFs. A constrained nonlinear optimization problem is constructed based on the collision-free path and the sigmoid curves. Finally, an optimal path of satisfying the constraints of collision avoidance and vehicle dynamics can be achieved by solving with the interior point algorithm. The main contributions of this study include:(1)A novel hybrid path planning method is proposed to get better collision-free path for improvements on vehicle stability and ride comfort during autonomous driving by combining potential field with sigmoid curve.(2)Based on the distribution function of two-dimensional joint probability density, an improved potential field of the obstacle is designed to mimic more realistic distribution of collision risk by decoupling the PF in longitudinal and lateral directions.(3)With the designed objective of the shortest path generation, the trajectory is optimized to improve the vehicle stability and the ride comfort during autonomous driving by considering the constraints of collision avoidance and vehicle dynamics.

This paper is organized as follows: Section 2 designs a potential field-based path planning and shows how PFBM can generate unexpected paths. Section 3 introduces the proposed HPFSM. Section 4 presents the validation and evaluation results by comparing the proposed method with PFBM both in the static and dynamic driving scenarios. Finally, Section 5 presents our conclusions.

## 2. An Improved Potential Field-Based Path Planning

Because of the good collision-free performance, PFBM has been widely used as a path planning approach for AVs [20]. The path planning process of PFBM can be mainly divided into two parts, namely, the part for designing PF functions and the collision-free route generation part for obstacle avoidance.

### 2.1. The Design of PF Functions

The PF is affected by obstacle properties including the obstacle’s physical characteristics (e.g., geometric shape and structure), and its dangerous degree is affected by the mass and motion state of the obstacle [21,22].

#### 2.1.1. Road PFs

Road PFs include the PF of road boundary and the PF of target lane. Since the potential field-based path planning is likely to fall into the local minima, especially in an unknown environment [23]. To avoid the local minima problem in our proposed path planning method, both the driving environment and the obstacles (vehicles) are assumed to be known, thus the potential fields can be designed and established more appropriately. In real intelligent transportation systems, these information can be obtained via the vehicle-to-vehicle and vehicle-to-infrastructure technologies [24]. Besides, we design the attractive potential field with the center line of the target lane to attract the ego vehicle driving to the target lane instead of with only one target point [25,26], which will reduce the chance to trap into the local minima by solving a series of optimization subproblems. In this study, the attractive PF is defined as:(1)$$U_{\mathrm{TrgL}}=a{\left(Y-Y_{\mathrm{TrgL}}\right)}^{2}.$$

The road boundary is designed as a repulsive PF in Equation (Equation 2), which is used for preventing the ego vehicle driving out of the road.
(2)$$U_{\mathrm{RBd}}=\left\{\begin{array}{c} b{\left(Y-Y_{\mathrm{Br}}\right)}^{2}\;Y\leq Y_{\mathrm{Br}}\\ \;b{\left(Y_{\mathrm{Bl}}-Y\right)}^{2}\;Y\geq Y_{\mathrm{Bl}}\\ 0\;Y\in \left(Y_{\mathrm{Br}},Y_{\mathrm{Bl}}\right) \end{array}\right.,$$
where a,b∈R, $Y_{\mathrm{Br}}<Y_{\mathrm{Bl}}$ and $Y_{\mathrm{TrgL}}\in \left(Y_{\mathrm{Br}},Y_{\mathrm{Bl}}\right)$; *a* and *b* respectively denote the shape coefficients of the target-lane PF and the road-boundary PF, which are used to adjust the amplitude of PF; *Y* denotes the lateral position of the ego vehicle; $Y_{\mathrm{TrgL}}$ denotes the lateral position of the center line of the target lane; $Y_{\mathrm{Bl}}$ and $Y_{\mathrm{Br}}$ denote the lateral positions of the left and right road boundaries, respectively.

Two examples of the successful applications without the local minima problem are shown in Figure 2. In the situation without obstacle vehicles, Figure 2a shows that the ego vehicle will always drive along the path (the central line of the target lane) with the minimum potential field when using our proposed method. In the situation with an obstacle vehicle in the target lane, Figure 2b shows that the planned path with minimum PF will lead the ego vehicle to overtake the obstacle vehicle and then drive back to the target lane.

#### 2.1.2. Obstacle Potential Field

The obstacle potential field (OPF) function is defined to construct the repulsive PF according to the longitudinal and lateral safe distances. The calculations of the safe distances are based on the relative speed (between the ego vehicle and the obstacle) and the maximum longitudinal/lateral deceleration of the ego vehicle [12], which means the velocities of the ego vehicle and the obstacle are required. The longitudinal and lateral safe distances $(X_{s}\left(t\right)$, $Y_{s}\left(t\right))$ are calculated as:(3)$$\left\{\begin{array}{c} X_{s}\left(t\right)=\frac{X_{o}}{2}+\frac{{\left(V_{x}\left(t\right)-V_{\mathrm{obs},x}\left(t\right)\right)}^{2}}{2a_{x,\max}\left(t\right)}\\ Y_{s}\left(t\right)=\frac{Y_{o}}{2}+\frac{{\left(V_{y}\left(t\right)-V_{\mathrm{obs},y}\left(t\right)\right)}^{2}}{2a_{y,\max}\left(t\right)} \end{array}\right.,$$
where $a_{x,\max}\left(t\right)\neq 0$ and $a_{y,\max}\left(t\right)\neq 0$ are the maximum longitudinal and lateral decelerations of the ego vehicle; $X_{o}$ and $Y_{o}$ are the length and width of the obstacle, respectively; $V_{\mathrm{obs},x}\left(t\right)$ and $V_{\mathrm{obs},y}\left(t\right)$ represent the longitudinal and lateral velocities of the obstacle, respectively; $V_{x}\left(t\right)$ and $V_{y}\left(t\right)$ are the longitudinal and lateral velocities of the ego vehicle, respectively.

The OPF can be decomposed along the longitudinal and lateral directions of the road coordinate system, and the definition domains of the two directions are usually independent and different [18]. Considering the above characteristics, a two-dimensional (2D) joint probability density distribution function is used as the basic function to define the OPF as:(4)$$U_{\mathrm{OPF}}\left(\ell ,\mu ,\Sigma \right)=\frac{a_{\mathrm{sta}}}{2\pi \sqrt{\left|\Sigma \right|}}e^{\left(-\frac{1}{2}{\left(\ell -\mu \right)}^{T}\Sigma^{-1}\left(\ell -\mu \right)\right)},$$
where
$$\mu ={\left(X_{\mathrm{obs}}\left(t\right),Y_{\mathrm{obs}}\left(t\right)\right)}^{T},\Sigma =\left[\begin{array}{cc} {X_{s}}^{2}\left(t\right) & 0\\ 0 & {Y_{s}}^{2}\left(t\right) \end{array}\right],\ell ={\left(X(t),Y(t)\right)}^{T},$$
where μ and Σ denote the mean and covariance matrix; $\left(X_{\mathrm{obs}}\left(t\right),Y_{\mathrm{obs}}\left(t\right)\right)$ and $\left(X(t),Y(t)\right)$ denote the positions of the obstacle and the ego vehicle at time *t*, respectively; $a_{\mathrm{sta}}\in R$ is the shape coefficient used to adjust the amplitude of OPF; $X_{s}\left(t\right)$ and $Y_{s}\left(t\right)$ are the calculated safe distances along the longitudinal and lateral directions of the road coordinate system at time *t*, respectively. Figure 3 is shown to illustrate that the OPF is adaptive to the safe distance, i.e., the OPF will vary with the velocities of the ego vehicle and the obstacles.

### 2.2. Collision-Free Path Generation

The idea behind PFBM is to generate a collision-free path occupied the minimum PF along the driving direction. The related attractive and repulsive PFs can be constructed and integrated according to the parameters described in Table 1.

The fused PF is shown in Figure 4. Based on this, the minimum PF path is obtained along the longitudinal direction of the road coordinate (*X* direction). Obviously, the generated collision-free path (the blue trajectory) is optimal subject to the defined PFs.

However, the path obtained using PFBM is not always smooth and expected, especially in these driving scenarios with multiple closely distributed obstacles. The blue path in Figure 5 shows the trajectory planned using PFBM [27] in a driving scenario with two closely distributed obstacles. Although the planned path is collision-free, it involves undesired driving maneuvering, which will affect the efficiency of obstacle avoidance and the tracking performance. An expected trajectory to mimic the real driver (e.g., the red route) is required for AV path planning to ensure the efficiency of the obstacle avoidance and to improve tracking performance.

The unexpected maneuvers of the planned path are more evident when the number of obstacles increases, as shown in Figure 6. In practical applications, the AVs would result in frequent unnecessary steering maneuvers when tracking this unexpected path. The planned paths in Figure 5 and Figure 6 indicate that the PFBMs are disadvantaged to be applied in the driving scenarios with multiple closely distributed obstacles. Therefore, a novel hybrid path planning method is proposed, which combines PFs with sigmoid curves to solve the problem of generating unexpected trajectory.

## 3. A Hybrid Path Planning Method

Based on the deterministic curve models, e.g., splines [28], clothoid curves [14], and polynomials [29], etc, the smooth candidate routes can be generated quickly and efficiently. However, it is difficult to shape desiring driving trajectory using the B-spline, C-spline, clothoid and even quintic polynomial models because of the strong coupling relationship among the tunable parameters. Considering the tunable feature of the parameters in the sigmoid curve model [30], which the amplitude, slope, and central symmetry point of the sigmoid curve can be adjusted independently. Therefore, a hybrid path planning method is proposed by combining the PFBM with sigmoid curves to obtain a smooth collision-free and efficient expected route.

### 3.1. Definition of the Sigmoid Curve

The process of obstacle avoidance is similar to that of vehicle lane change. The trajectory is tangent to the center lines of the related lanes at the start and end points according to the standard lane change path [31]. In this paper, the sigmoid function is introduced as an essential function for generating obstacle avoidance paths. The definition is presented in Equation (Equation 5):(5)$$\left\{\begin{array}{c} f_{\mathrm{sig}}\left(x\right)=P_{b}\cdot sigmoid\left(x,P_{c},P_{a}\right)\\ sigmoid\left(x,P_{c},P_{a}\right)=\frac{1}{1+e^{-P_{a}\left(x-P_{c}\right)}} \end{array}\right.,$$
where $P_{a}$, $P_{b}$ and $P_{c}$ are the related parameters to shape the sigmoid curve. The parameter $P_{a}$ represents the maximum slope, $P_{b}$ is the amplitude coefficient, $P_{c}$ denotes the centrosymmetric point and the point of maximum slope. These parameters can be used to determine a sigmoid curve uniquely.

### 3.2. Tunable Features of the Sigmoid Curve

Figure 7 shows the tunable features of the sigmoid curve including the maximum slop, the amplitude and the central point. Figure 7a shows that the centrosymmetric point can be adjusted independently using the parameter of $P_{c}$, which can be used to move the sigmoid curve in longitudinal direction. Figure 7b shows that the amplitude can be adjusted using the parameter of $P_{b}$, which can be applied to compress or stretch the sigmoid curve in the lateral direction. Furthermore, Figure 7c indicates that the maximum slope is also independently tunable using the parameter of $P_{a}$, which can be used to adjust the maximum slope of the sigmoid curve at the centrosymmetric point.

### 3.3. Configuration of the Sigmoid Curve

#### 3.3.1. Collision-Free Path Generation of PFBM

The potential fields are integrated according to Equation (Equation 6):(6)$$U_{\mathrm{PF}}\left(t\right)=U_{\mathrm{TargL}}\left(t\right)+U_{\mathrm{RBd}}\left(t\right)+U_{\mathrm{OPF}}\left(t\right),$$
where $U_{\mathrm{PF}}\left(t\right)$ is the integrated PF, $U_{\mathrm{TargL}}\left(t\right)$, $U_{\mathrm{RBd}}\left(t\right)$ and $U_{\mathrm{OPF}}\left(t\right)$ are the corresponding target lane PF, the road boundary PF and the obstacle PF at time *t*, respectively.

The PFBM collision-free path is obtained through Equation (Equation 7):(7)$$\left\{X_{\min},Y_{\min}\right\}=\min_{\left\{x_{\min}\left(t\right),y_{\min}\left(t\right)\right\}}U_{\mathrm{PF}}\left(t\right),$$
where $\left\{X_{\min},Y_{\min}\right\}$ represent the collision-free path with the minimum PF along the longitudinal direction.

The corresponding lateral positions of the obstacles mapping to the collision-free path are obtained by interpolation through Equation (Equation 8):(8)$$\left\{\begin{array}{c} \forall :x_{\mathrm{obs}}^{j}\in \left[x_{i},x_{i+1}\right],j\in M\\ \exists :y_{\mathrm{obs}}^{j}=interp1\left(X_{\min},Y_{\min},x_{\mathrm{obs}}^{j}\right) \end{array}\right.\Rightarrow y_{\mathrm{obs}}^{j}=\frac{y_{i+1}-y_{i}}{x_{i+1}-x_{i}}\left(x_{\mathrm{obs}}^{j}-x_{i}\right)+y_{i},$$
where $\left(x_{i},y_{i}\right)\in \left\{X_{\min},Y_{\min}\right\}$ and $\left(x_{i+1},y_{i+1}\right)\in \left\{X_{\min},Y_{\min}\right\}$ are two known waypoints in the collision-free path; interp1 denotes the one-dimensional linear interpolation function for calculating the corresponding lateral coordinates to the collision-free path; *i* and *j* denote the index of the waypoints and the index of the obstacles, respectively; *M* is the amount of the obstacle; $x_{\mathrm{obs}}^{j}$ and $y_{\mathrm{obs}}^{j}$ are the longitudinal and lateral coordinates corresponding to the PFBM path, respectively.

The planned collision-free path is composed of several sigmoid curves, and the definition domains are varying with the positions of obstacles. The definition domains of the sigmoid curves are determined using Equation (Equation 9):(9)$$\Omega_{x,i}=\left\{\begin{array}{c} {}[x_{\mathrm{start}},x_{\mathrm{obs}}^{i}]\;i=1\\ \left[x_{\mathrm{obs}}^{i-1},x_{\mathrm{obs}}^{i}\right]\;i\in \left(1,n\right)\\ \left[x_{\mathrm{obs}}^{i-1},x_{\mathrm{end}}\right]\;i=n \end{array}\right.,$$
where $\Omega_{x,i}$ denotes the definition domain of the *i*th sigmoid curve, $x_{\mathrm{start}}$ and $x_{\mathrm{end}}$ indicate the start and end points of the planning horizon, n=M+1 represents the amount of sigmoid curves. Considering the detection ranges of on-board sensors [32], the planning horizon is limited to 200 m.

#### 3.3.2. Parameter Configuration

Some key way-points of the collision-free path can be obtained using Equations (Equation 7) and (Equation 8). Since the amplitude of sigmoid curve is related to the lateral coordinates of the target lane $\left(Y_{\mathrm{TrgL}}\right)$ and the key waypoints $\left(y_{\mathrm{obs}}\right)$, the amplitude of the curve is determined in Equation (Equation 10):(10)$$P_{b,i}=\left\{\begin{array}{c} y_{\mathrm{obs}}^{i}-Y_{\mathrm{TrgL}}\;i=1\\ y_{\mathrm{obs}}^{i}-y_{\mathrm{obs}}^{i-1}\;i\in \left(1,n\right)\\ Y_{\mathrm{TrgL}}-y_{\mathrm{obs}}^{i-1}\;i=n \end{array}\right..$$

The slope parameter can be determined using Equation (Equation 11):(11)$$P_{a,i}=k_{1,i}\cdot sign\left(P_{b,i}\right),$$
where $k_{1,i}$ denotes the maximum slope at the centrosymmetric point and sign is the sign function.

The centrosymmetric point of the sigmoid curve is defined in Equation (Equation 12):(12)$$P_{c,i}=\left\{\begin{array}{c} x_{\mathrm{obs}}^{i}-k_{2,i}X_{s}\;i=1\\ x_{\mathrm{obs}}^{i-1}+\frac{k_{2,i}\left(x_{\mathrm{obs}}^{i}-x_{\mathrm{obs}}^{i-1}\right)}{2}\;i\in \left(1,n\right)\\ x_{\mathrm{obs}}^{i-1}+k_{2,i}X_{s}\;i=n \end{array}\right.,$$
where $k_{2,i}\geq 1$ is a tunable coefficient.

When the above three parameters and the bias have been obtained in the definition domain $\Omega_{x,i}$, the sigmoid curve can be determined uniquely using Equation (Equation 13):(13)$$f_{sig,i}=P_{b,i}\cdot sigmoid\left(x,P_{c,i},P_{a,i}\right)+b_{i}\;,$$
$$x\in \Omega_{x,i},\;b_{i}=\left\{\begin{array}{c} Y_{\mathrm{TrgL}}\;i=1\\ y_{\mathrm{obs}}^{i-1}\;i\neq 1 \end{array}\right.,$$
where $f_{sig,i}$ denotes the *i*th sigmoid curve function, $b_{i}$ is the corresponding bias. Figure 8 shows the parameters that shape the sigmoid curve in detail.

### 3.4. Trajectory Optimization with Sigmoid Curves

Since the coefficients of $k_{1,i}$ and $k_{2,i}$ are not determined yet, a series of sigmoid curves can be generated by the above configurations. An optimization objective function is designed to obtain the shortest path subject to the constraints of the lateral acceleration and the yaw rate to ensure collision avoidance and to improve the vehicle stability of the autonomous driving. The distance of sigmoid curve is calculated according to Equation (Equation 14):(14)$$S_{sig,i}=\int_{x_{\mathrm{start}}^{i}}^{x_{\mathrm{end}}^{i}}\sqrt{1+{\dot{f}}_{sig,i}^{2}\left(x\right)}dx.$$

#### 3.4.1. Collision Avoidance Constraint

The planned path generated by PFBM is collision-free, which can be used as the constraints of collision avoidance to assist configuring the collision-free sigmoid curves. The collision-free feature can be determined if the sigmoid curves are always farther to the obstacle than that of the collision-free path of PFBM. As Figure 9a indicates, the collision feature cannot be deduced directly and an additional check is required. Therefore, the constraints of collision avoidance should be considered to ensure the collision-free feature of the candidate sigmoid curves.

Instead of combining the geometric information of obstacles [33], the collision avoidance can be ensured by comparing the lateral positions of the candidate paths to that of the PFBM. As illustrated in Figure 9b, collision avoidance is ensured when the red line is completely above the blue line under this situation.

The constraints to ensure collision avoidance based on the path of PFBM are shown in Equation (Equation 15):(15)$$K_{Ineq,i}^{Cons}\left(1\right):=\left\{\begin{array}{c} f_{sig,i}\left(x_{i}\right)\geq Y_{\min},\;y_{\mathrm{obs}}^{i}\geq Y_{\mathrm{obs}}\\ f_{sig,i}\left(x_{i}\right)\leq Y_{\min},\;y_{\mathrm{obs}}^{i}<Y_{\mathrm{obs}} \end{array}\right.,$$
where $x_{i}\in \Omega_{x,i}$ denotes the longitudinal range of the road, and $K_{Ineq,i}^{Cons}$ denotes the inequality constraints of $k_{1,i}$ and $k_{2,i}$.

#### 3.4.2. The Constraints of Vehicle Dynamics

The constraints of vehicle dynamics should also be considered in path planning module to improve the vehicle stability [34,35] during path tracking. The vehicle stability and ride comfort can be well evaluated based on the lateral acceleration and yaw rate during the path tracking. Assuming that the target velocity is invariant during path tracking, the yaw rate is considered as shown in Equation (Equation 16):(16)$$\left\{\begin{array}{c} \omega_{v,i}=\rho_{i}V\\ |\omega_{v,i}|\leq \omega_{s} \end{array}\right.,$$
where $\omega_{v,i}$ (rad/s) is the yaw rate of the *i*th curve at a speed of *V* (m/s), $\rho_{i}$ is the curvature of the *i*th curve and $\omega_{s}$ (rad/s) denotes the yaw rate constraint to ensure path tracking stability.

The lateral acceleration is considered as follows in Equation (Equation 17):(17)$$\left\{\begin{array}{c} a_{y,i}=Vcos\theta_{i}\omega_{v,i}\\ |a_{y,i})|\leq a_{s} \end{array}\right.,$$
where *V* (m/s) denotes the target speed for path tracking, $a_{s}$ (m/s${}^{2}$) denotes the constraints of lateral acceleration in path planning module, $a_{y,i}$ is the lateral acceleration of the *i*th curve.

The constraints of vehicle dynamics can be transformed into a constraint of path curvature in Equation (Equation 18):(18)$$K_{Ineq,i}^{Cons}\left(2\right):=\left\{\begin{array}{c} \rho_{i}=\frac{\left|{\ddot{f}}_{sig,i}\left(x\right)\right|}{{\left(1+{\dot{f}}_{sig,i}{\left(x\right)}^{2}\right)}^{3/2}}\\ |\rho_{i}|\leq \rho_{\cos}\\ \rho_{\cos}=min\left(\frac{a_{s}}{V^{2}},\frac{w_{s}}{V}\right) \end{array}\right.,$$
where $\rho_{\cos}$ denotes the curvature constraint of planned path considering the ride comfort and vehicle stability in path planning module.

#### 3.4.3. Geometric Constraints

The geometric constraints include amplitude, start point, endpoint and central symmetric point constraints. The end point constraint is defined as:(19)$$K_{Eq,i}^{Cons}\left(1\right):=\left\{\begin{array}{c} x_{i}^{\mathrm{end}}=\Omega_{x,i}\left(end\right)\\ y_{i}^{\mathrm{end}}=interp1\left(X_{\min},Y_{\min},x_{i}^{\mathrm{end}}\right)\\ f_{sig,i}\left(x_{i}^{\mathrm{end}}\right)=y_{i}^{\mathrm{end}}\\ 0<{\dot{f}}_{sig,i}{|}_{x_{i}^{\mathrm{end}}}\leq \epsilon \end{array}\right.,$$
where ϵ denotes the infinitesimal value, and $K_{Eq,i}^{Cons}$ denotes the equality constraints of $k_{1,i}$ and $k_{2,i}$.

The start point constraint is defined as:(20)$$K_{Eq,i}^{Cons}\left(2\right):=\left\{\begin{array}{c} x_{i}^{\mathrm{start}}=\Omega_{x,i}\left(start\right)\\ y_{i}^{\mathrm{end}}-f_{sig,i}\left(x_{i}^{\mathrm{start}}\right)=P_{b,i}\\ 0<{\dot{f}}_{sig,i}{|}_{x_{i}^{\mathrm{start}}}\leq \epsilon \end{array}\right..$$

The constraint of the centrosymmetric point $P_{c,i}$ is defined as:(21)$$K_{Ineq,i}^{Cons}\left(3\right):=\left\{\begin{array}{c} x_{i}^{\mathrm{end}}-P_{c,i}\geq X_{s}\\ x_{i}^{\mathrm{start}}<P_{c,i}<x_{i}^{\mathrm{end}} \end{array}\right..$$

The inequality constraints are thus summarized as:(22)$$K_{\mathrm{Ineq}}^{i}=\left\{K_{Ineq,i}^{Cons}\left(1\right),K_{Ineq,i}^{Cons}\left(2\right),K_{Ineq,i}^{Cons}\left(3\right)\right\},$$
where $K_{Ineq}^{i}$ refers to the inequality constraints of the *i*th curve, including the constraints of the collision avoidance, the constraints of the lateral acceleration, the constraints of the yaw rate and the constraints of the geometric.

The constrained nonlinear optimization problem is formulated as:(23)$$\min_{\left\{k_{1,i},k_{2,i}\right\}}\int_{x_{\mathrm{start}}^{i}}^{x_{\mathrm{end}}^{i}}\sqrt{1+{\dot{f}}_{sig,i}^{2}\left(x\right)}dx$$
$$\begin{array}{c} \;\;s.t.\\ \;\;\;x_{i}^{\mathrm{end}}=\Omega_{x,i}\left(end\right)\\ \;\;\;y_{i}^{\mathrm{end}}=interp1\left(X_{\min},Y_{\min},x_{i}^{\mathrm{end}}\right)\\ \;\;\;f_{sig,i}\left(x_{i}^{\mathrm{end}}\right)=y_{i}^{\mathrm{end}}\\ \;\;\;x_{i}^{\mathrm{start}}=\Omega_{x,i}\left(start\right)\\ \;\;\;y_{i}^{\mathrm{end}}-f_{sig,i}\left(x_{i}^{\mathrm{start}}\right)=P_{b,i}\\ \;\;\;0<{\dot{f}}_{sig,i}{|}_{x_{i}^{\mathrm{end}}}\leq \epsilon \\ \;\;\;0<{\dot{f}}_{sig,i}{|}_{x_{i}^{\mathrm{start}}}\leq \epsilon ,\epsilon >0\\ \;\;\;\left\{k_{1,i},k_{2,i}\right\}\in K_{\mathrm{Ineq}}^{i} \end{array}$$

A driving scenario with one static obstacle is proposed to analyze the planned path of HPFSM. The relevant parameters for realizing HPFSM are described in Table 2. Figure 10 shows that there are several trajectories satisfying the collision-free constraints, e.g., the blue solid and dotted curves. With a target speed of 20 m/s, the optimal trajectory among the candidate curves is the shortest trajectory (composed with the red and black curves) satisfying the constraints of vehicle dynamics, which require the yaw rate and lateral acceleration are within 25 deg/s and 2 m/s${}^{2}$, respectively.

Figure 11a shows the curvature of the optimal trajectory generated using HPFSM. It shows the curvature is continuous, which means the optimal trajectory is drivable. Figure 11b shows the yaw rate and lateral acceleration calculated under the target tracking speed of 20 m/s. It indicates that both the constraints of the lateral acceleration and the yaw rate are effectiveness during the path planning. The results of Figure 11 illustrate that the trajectory can be optimized to satisfy the constraints of vehicle dynamics with the parameters optimization of sigmoid curves.

## 4. Verification and Discussion

To further examine and evaluate the proposed approach, a static and a dynamic driving scenario are designed for simulation, respectively. The parked vehicles are considered as the obstacles in the static scenario, and vehicles with short intervals are introduced as the overtaking objects in the dynamic scenario.

### 4.1. Driving Scenarios for Simulation and Evaluation

A static scenario is with three cars parked on the roadsides as shown in Figure 12. There are two parked cars located in the target lane, while the ego vehicle is approaching to the parked car with a speed of 20 m/s. The positions of the three parked vehicles are $(X_{\mathrm{obs},1}=80$ m, $Y_{\mathrm{obs},1}=1.5$ m ), $(X_{\mathrm{obs},2}=180$ m, $Y_{\mathrm{obs},2}=6.2$ m) and $(X_{\mathrm{obs},3}=280$ m, $Y_{\mathrm{obs},3}=1.5$ m), respectively. The constraints of the lateral acceleration and yaw rate are designed within 2 m/s${}^{2}$ and 25 deg/s, respectively.

A dynamic overtaking scenario is designed with three leading vehicles located with short distances, as shown in Figure 13. The red dotted line denotes the target lateral position, i.e., the center line of the target lane. Three leading vehicles are driving with a constant speed ($V_{\mathrm{obs},1}=V_{\mathrm{obs},2}=V_{\mathrm{obs},3}=15$ m/s) from different initial positions ($X_{\mathrm{obs},1}=50$ m, $X_{\mathrm{obs},2}=70$ m, $X_{\mathrm{obs},3}=85$ m). The initial position and speed of the ego vehicle are set as $X_{\mathrm{ego}}=0$ m and $V_{\mathrm{ego}}=15$ m/s, respectively. The target speed and lateral position of the ego vehicle are set as 20 m/s and 1.75 m, respectively. Meanwhile, the constraints of yaw rate and lateral acceleration are designed within 2 m/s${}^{2}$ and 25 deg/s, respectively.

### 4.2. Path Tracking Controller for Validation

Since the main purpose of path planning is to provide an expected reference trajectory for path tracking, it is more meaningful to evaluate the proposed path planning method with the combination of a path tracking controller. Based on these, a linear time-varying model predictive tracking controller (LTV-MPC) [20] is used to evaluate the HPFSM by comparing with PFBM. The 3-DOF bicycle model, including the longitudinal, lateral and yaw directions, is used as the prediction model of the LTV-MPC.

The dynamics equations of the 3-DOF dynamics model are presented in Equation (Equation 24):(24)$$\left\{\begin{array}{c} m\left({\dot{v}}_{x}-\omega v_{y}\right)=F_{x}\cos\delta \\ m\left({\dot{v}}_{y}+\omega v_{x}\right)=F_{y,r}+F_{y,f}\cos\delta \\ I_{z}\dot{\omega }=F_{y,f}L_{f}\cos\delta -F_{y,r}L_{r} \end{array}\right..$$

The motion equations of the vehicle are shown in Equation (Equation 25):(25)$$\left\{\begin{array}{c} \dot{X}=v_{x}\cos\varphi -v_{y}\sin\varphi \\ \dot{Y}=v_{x}\sin\varphi +v_{y}\cos\varphi \end{array}\right.,$$
where $v_{x}$, $v_{y}$ and ω are the longitudinal velocity, lateral velocity and yaw rate of the vehicle, respectively; *X*, *Y* and φ denote the vehicle longitudinal, lateral position and the heading angle; *m* and $F_{x}$ represent the vehicle mass and longitudinal force of the front-driving tire; $F_{y,f}$ and $F_{y,r}$ denote the lateral force of front and rear tires; $L_{f}$, $L_{r}$ and $I_{z}$ represent the front, rear wheelbase and the vehicle inertia around vertical axis, respectively; δ is the steering angle of the front wheel. The relevant vehicle parameters are the same as Table I in [17]. The configuration parameters of the LTV-MPC are presented in Table 3.

### 4.3. Results and Discussion

#### 4.3.1. Static Scenario

The comparisons of trajectories between HPFSM and PFBM are shown in Figure 14a, respectively. The green and red trajectories are the generated paths of PFBM and HPFSM, respectively. It shows that the optimized path by HPFSM is more feasible to be an expected driving trajectory, because the path is smoother than that of PFBM without increasing in length (400.87 m vs. 403.88 m). The shaded part illustrates that the red trajectory is collision-free by comparing the lateral positions of the two paths. The comparison of the curvatures between the two trajectories is shown in Figure 14b, which further illustrates the red path is smoother than the green one.

The planned paths of HPFSM and PFBM are tracked by the LTV-MPC with a target speed of 20 m/s, respectively. The comparisons of the yaw rate and lateral acceleration between the two methods are shown in Figure 15. The comparison of lateral acceleration shows that the instantaneous and average values of HPFSM are smaller than that of PFBM in Figure 15a, which illustrates the ride comfort is improved with HPFSM. Meanwhile, Figure 15b shows the comparison of yaw rate, which illustrates that the yaw rate based on PFBM does not satisfy the designed constraint of 25 deg/s; however, the yaw rate based on HPFSM can be well constrained. This implies that the stability of the ego vehicle is also better while tracking the path of HPFSM.

The improvements with HPFSM in the static scenario is analyzed in Table 4. It shows that the maximum and average lateral accelerations are decreased almost 60% (6.254 m/s${}^{2}$
vs. 2.504 m/s${}^{2}$) and 40.6% (0.475 m/s${}^{2}$
vs. 0.282 m/s${}^{2}$) comparing to PFBM, respectively. Meanwhile, the yaw rate is also optimized in the maximum and average values, respectively. The maximum and average yaw rates with HPFSM are improved 60.47% (44.17 deg/s vs. 17.459 deg/s) and 28.2% (3.517 deg/s vs. 2.524 deg/s), respectively.

#### 4.3.2. Dynamic Scenario

The tracking velocities of HPFSM and PFBM are shown in Figure 16a to ensure a consistent speed environment during the overtaking task. The trajectories of the two methods are shown in Figure 16b; they show that the ego vehicle can finish the overtaking task with both of these two methods. However, the trajectory with PFBM shows an sudden fluctuation at the position around X=300 m, which will result in sharp steering maneuvers as shown in Figure 16c. These unexpected steering maneuvers will further affect both the driving safety and stability.

The yaw rate and the lateral acceleration of the two methods are compared in Figure 17. The yaw rate is constrained within 10 deg/s while tracking the path of HPFSM; however, the yaw rate is beyond the designed constraint of 25 deg/s while tracking the path of PFBM, as shown in Figure 17a. Meanwhile, the comparison of the lateral acceleration in Figure 17b shows the lateral acceleration based on HPFSM is much smaller than that of PFBM during tracking. These illustrate that both the vehicle stability and the ride comfort of the ego vehicle are improved with the proposed HPFSM comparing to the PFBM. The improvements with HPFSM in the dynamic scenario are shown in Table 5. This indicates that the maximum and average lateral accelerations with HPFSM are decreased 87.8% (2.4 m/s${}^{2}$
vs. 0.29 m/s${}^{2}$) and 83.9% (0.18 m/s${}^{2}$
vs. 0.029 m/s${}^{2}$), respectively. Meanwhile, the yaw rate is also optimized in the maximum and average values compared to the PFBM. The maximum and average yaw rates are improved 82.8% (20.4 deg/s vs. 3.5 deg/s) and 72.2% (1.7 deg/s vs. 0.47 deg/s), respectively.

## 5. Conclusions

A hybrid path planning is proposed to achieve an expected path generation and to improve the vehicle stability and the ride comfort during autonomous driving by combining the potential field with the sigmoid curve. The collision avoidance and the vehicle dynamics are considered to obtain the shortest collision-free trajectory composed by sigmoid curves. The multiobstacle static and dynamic scenarios are designed to examine the effectiveness of HPFSM, respectively. To evaluate the performance of autonomous driving with HPFSM, an LTV-MPC is used to track the planned paths of HPFSM and PFBM, respectively. The simulation results of the static scenario show that the maximum and average lateral accelerations are decreased 60% and 40%, and the maximum and average yaw rates are decreased almost 60.47% and 28.2%, respectively. The results of the simulated dynamic scenario show the same trend as the static scenario with a decrease of almost 80% in the indexes of both the lateral acceleration and the yaw rate. However, these improvements are achieved on the basis of the analysis of the simulation results; the figures are likely to be more modest in the practical application. These simulation results indicate that the vehicle stability and the ride comfort are well improved with the proposed method during autonomous driving. How the local minima problem can be completely or sufficiently avoided in more complex and unknown driving scenarios with more traffic participants is still a challenging task and should be further addressed in the future work. Meanwhile, our future work will present experimental applications of the proposed method under real driving scenarios.

## Figures and Tables

**Figure 1 sensors-20-07197-f001:**
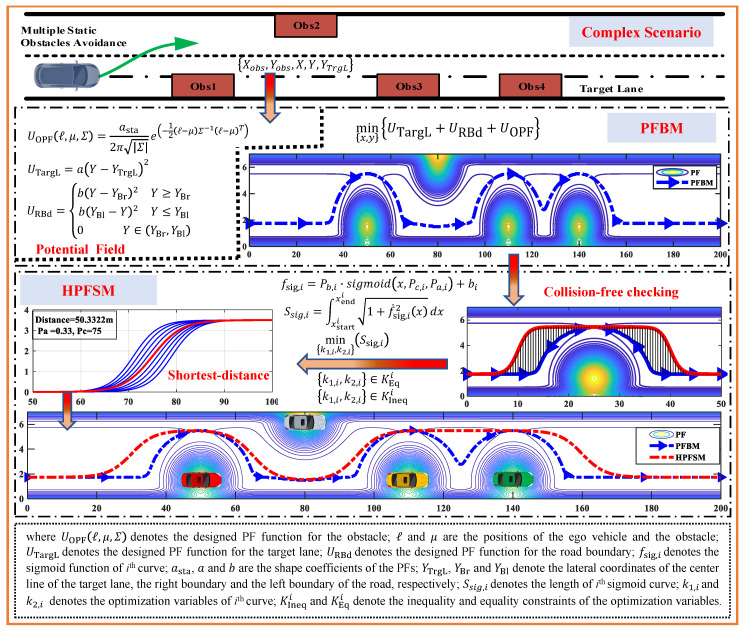
Framework of the hybrid path planning method by combining potential fields with sigmoid curves.

**Figure 2 sensors-20-07197-f002:**
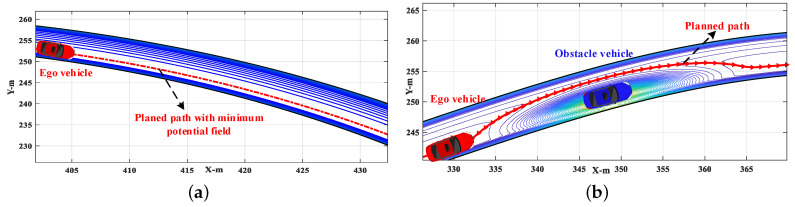
Examples without trapping into local minima: (**a**) scenario without obstacle vehicle; (**b**) scenario with an obstacle vehicle.

**Figure 3 sensors-20-07197-f003:**
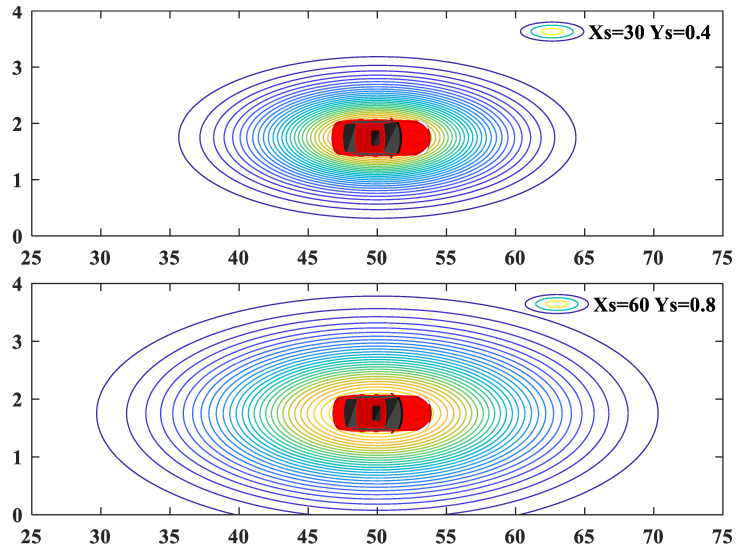
Potential fields with different longitudinal and lateral safe distances.

**Figure 4 sensors-20-07197-f004:**
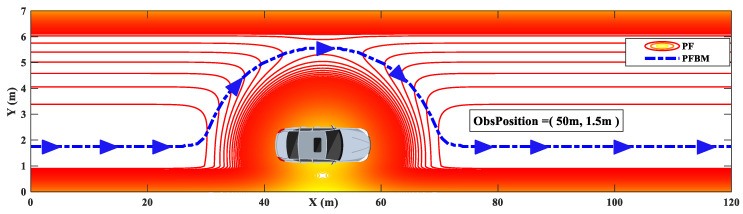
PFBM path planning for collision avoidance.

**Figure 5 sensors-20-07197-f005:**
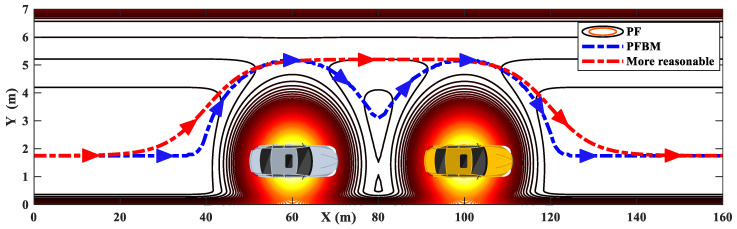
A driving scenario with two obstacle vehicles.

**Figure 6 sensors-20-07197-f006:**
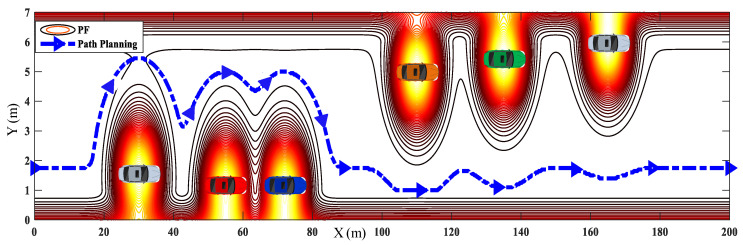
A driving scenario with multivehicle.

**Figure 7 sensors-20-07197-f007:**
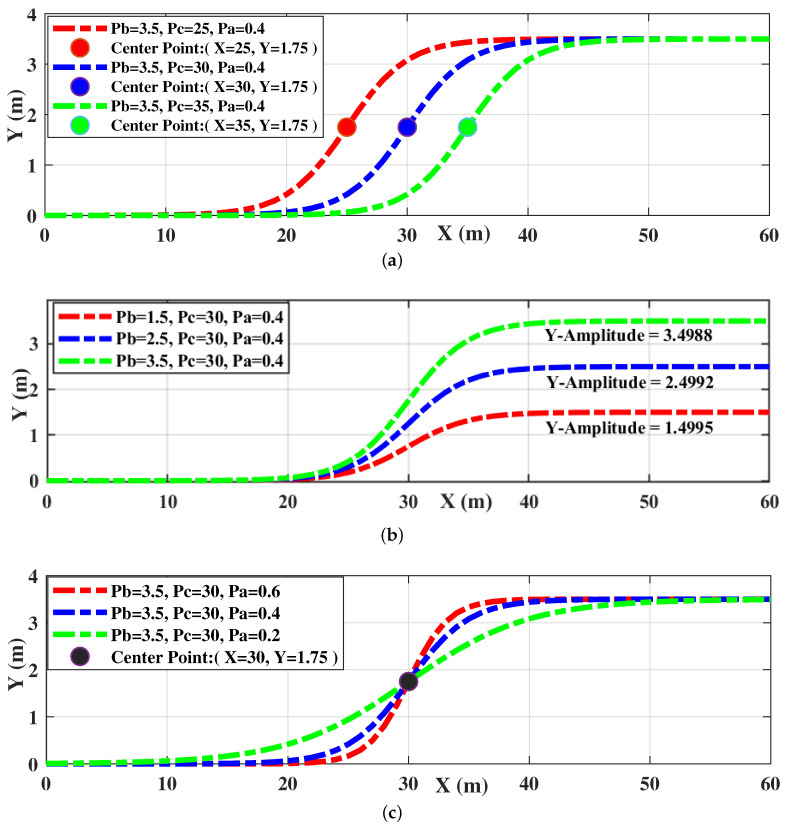
Tunable features of the sigmoid curve: (**a**) Sigmoid curves with different central point; (**b**) Sigmoid curves with different central point; (**c**) Sigmoid curves with different maximum slope.

**Figure 8 sensors-20-07197-f008:**
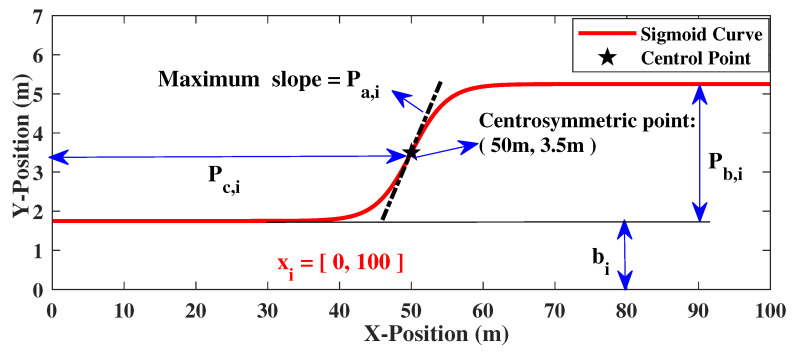
The parameters of sigmoid curve.

**Figure 9 sensors-20-07197-f009:**
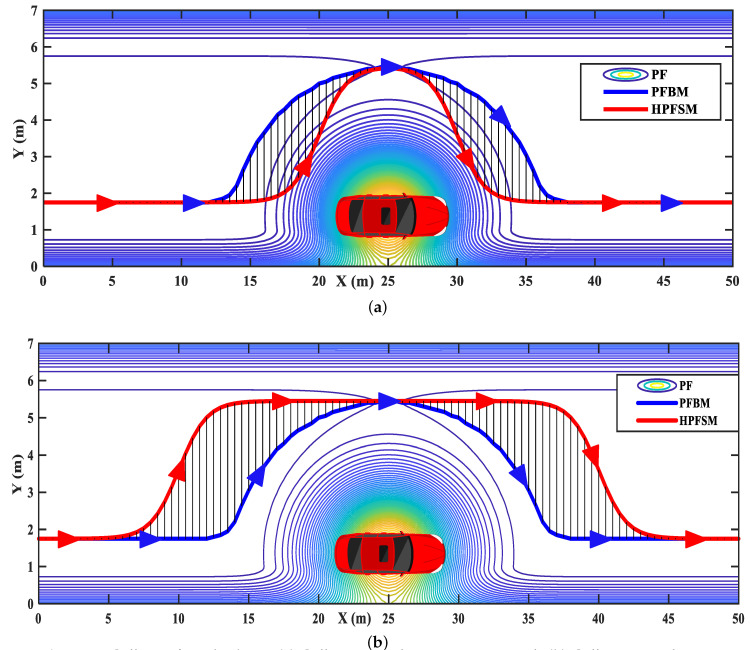
Collision-free checking: (**a**) Collision avoidance is not ensured; (**b**) Collision avoidance is ensured.

**Figure 10 sensors-20-07197-f010:**
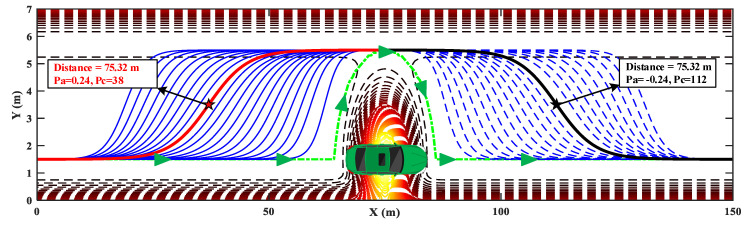
The optimal trajectory with sigmoid curves in a static scenario.

**Figure 11 sensors-20-07197-f011:**
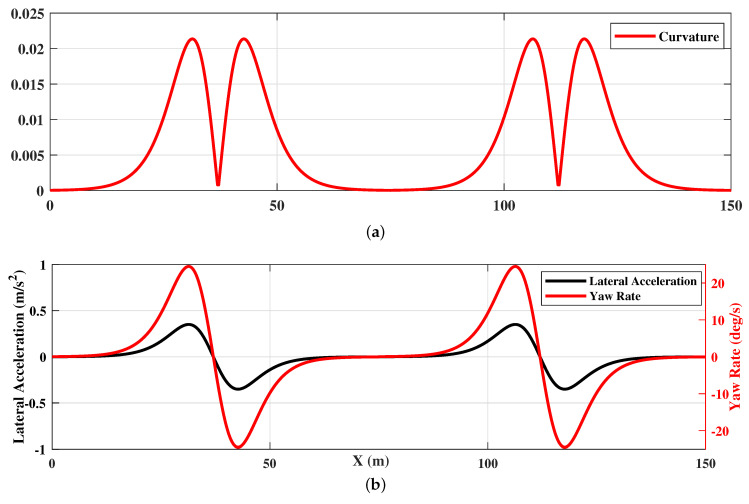
HPFSM path planning with constraints: (**a**) curvature of the planned trajectory; (**b**) yaw rate and lateral acceleration.

**Figure 12 sensors-20-07197-f012:**
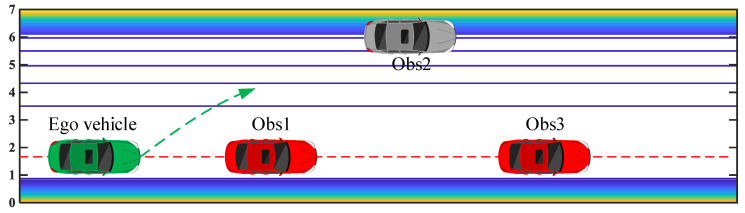
A static driving scenario with three vehicles parked on roadsides.

**Figure 13 sensors-20-07197-f013:**
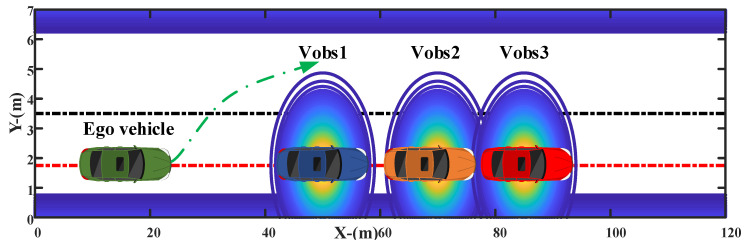
Leading vehicles driving with short interval distance.

**Figure 14 sensors-20-07197-f014:**
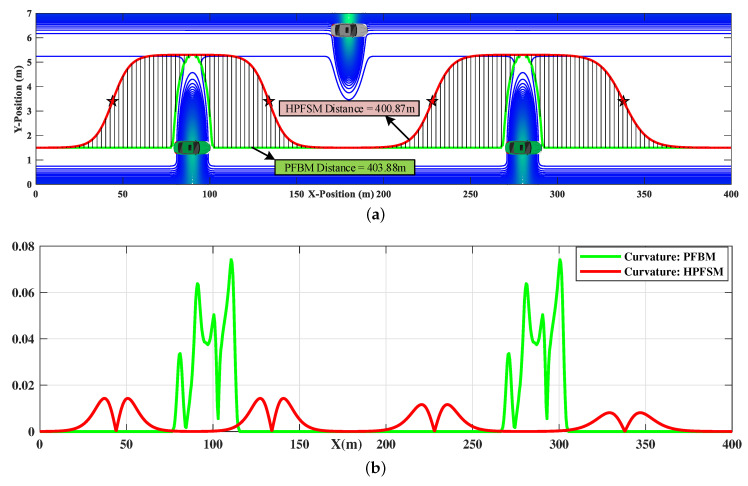
The path comparisons between HPFSM and PFBM in a static scenario: (**a**) Trajectory comparison: HPFSM vs PFBM; (**b**) Curvature comparison: HPFSM vs PFBM.

**Figure 15 sensors-20-07197-f015:**
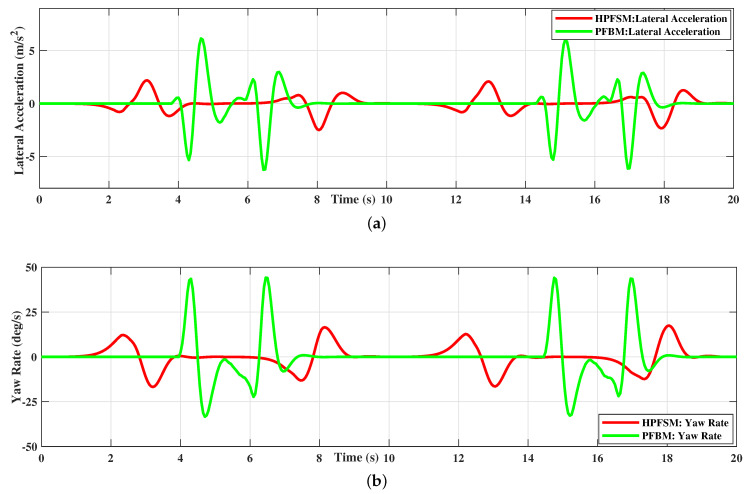
Path tracking comparisons between the two methods in a static scenario: (**a**) LTV-MPC tracking: Lateral acceleration comparison; (**b**) LTV-MPC tracking: Yaw rate comparison.

**Figure 16 sensors-20-07197-f016:**
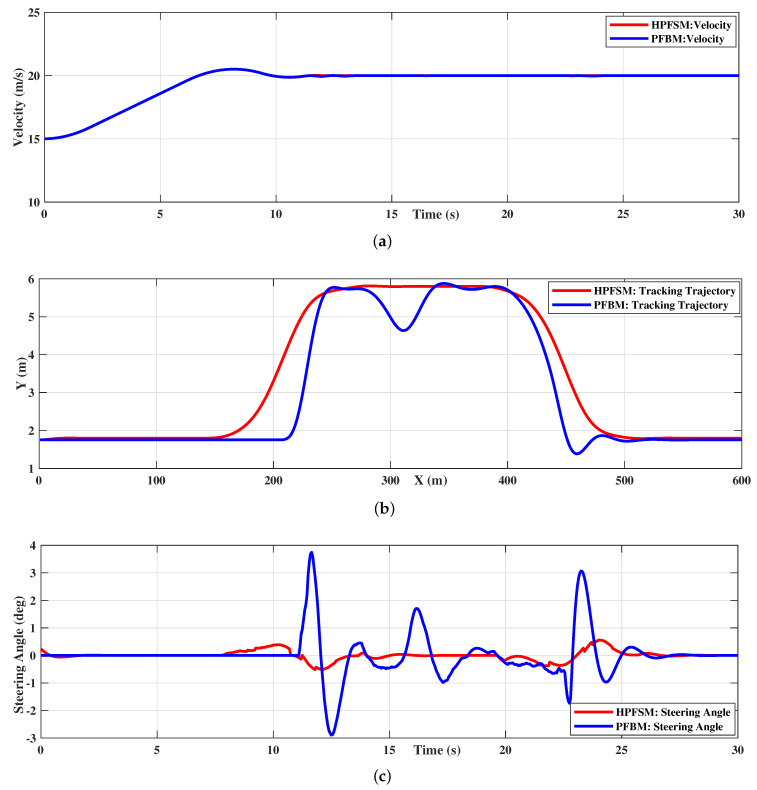
An overtaking scenario with multivehicle distributed in short interval distance: (**a**) HPFSM vs. PFBM: Tracking velocity; (**b**) HPFSM vs. PFBM: Tracking trajectory; (**c**) HPFSM vs. PFBM: Steering angle of front wheel.

**Figure 17 sensors-20-07197-f017:**
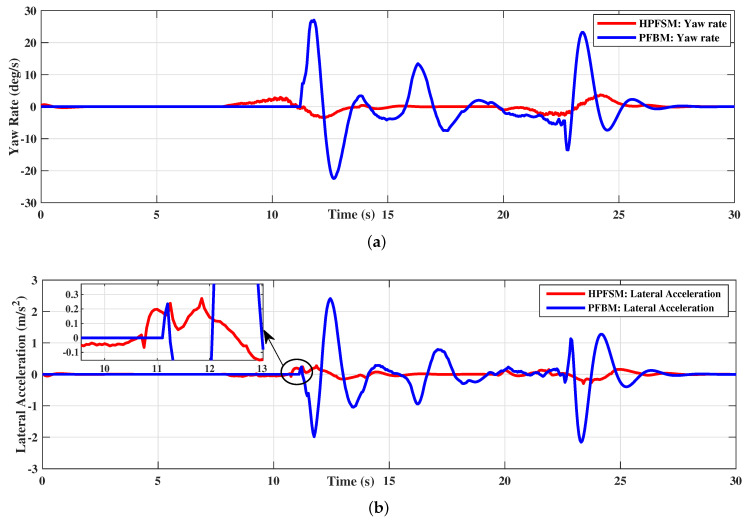
The comparisons of yaw rate and lateral acceleration: (**a**) HPFSM vs. PFBM: Yaw rate; (**b**) HPFSM vs. PFBM: Lateral acceleration.

**Table 1 sensors-20-07197-t001:** Parameters for PF construction.

Parameter	Value	Parameter	Value	Parameter	Value
*X* (m)	0∼200	$Y_{\mathrm{TrgL}}$ (m)	1.75	$Y_{\mathrm{obs}}$ (m)	1.5
*Y* (m)	0∼7	$Y_{\mathrm{Bl}}$ (m)	6	$X_{s}$ (m)	20
*a*	0.5	$Y_{\mathrm{Br}}$ (m)	1	$Y_{s}$ (m)	1.5
*b*	100	$X_{\mathrm{obs}}$ (m)	50	$a_{\mathrm{sta}}$	1 $\times 10^{4}$

**Table 2 sensors-20-07197-t002:** Initialization parameters.

Parameter	Value	Parameter	Value	Parameter	Value
μ	(75,1.5)	$b_{1}$	0	$x_{2}$	[75,150]
Σ	diag([10,1.5])	$P_{c,1}$	[0,75]	$b_{2}$	3.5
$P_{b,1}$	3.5	$P_{a,1}$	[0,1]	$P_{c,2}$	[75,150]
$x_{1}$	[0,75]	$P_{b,2}$	−3.5	$P_{a,2}$	[−1,0]
*V*	20 (m/s)	$a_{s}$	2 (m/s${}^{2}$)	$\omega_{s}$	25 (deg/s)

**Table 3 sensors-20-07197-t003:** Parameters of the MPC Controller.

Symbol	Description	Value [Units]
$N_{p}$	Prediction horizon	20 [unitless]
$N_{u}$	Control variable’s number	2 [unitless]
$N_{s}$	State variable’s number	6 [unitless]
$T_{s}$	Sampling period	0.05 [s]
$\delta_{w}$	Limitation of steering wheel angle	[−540,540] ${[}^{\circ }]$
$\Delta \delta_{w}$	Steering wheel angle rate	[−5,5] ${[}^{\circ }]$
$F_{x}$	Longitudinal tire force limitation	[−2000,2000] [N]
$\Delta F_{x}$	Tire force rate limitation	[−50,50] [N]
Q	Weights matrix of states tracking	diag($[1\times 10^{-7}$, $1\times 10^{2}$, $1\times 10^{-7}$, 0, $1\times 10^{-7}$, 0])
R	Weights matrix of control variables	$diag([1\times 10^{-7}$, $1\times 10^{-5}])$

**Table 4 sensors-20-07197-t004:** Results comparisons in the static scenario.

Symbol	Description	HPFSM	PFBM	−(%)
$a_{y,\max}$ (m/s${}^{2}$)	Maximum lateral acceleration	2.504	6.254	59.9
$a_{y,\mathrm{mean}}$ (m/s${}^{2}$)	Average lateral acceleration	0.282	0.475	40.6
$\omega_{\max}$ (deg/s)	Maximum yaw rate	17.459	44.170	60.47
$\omega_{\mathrm{mean}}$ (deg/s)	Average yaw rate	2.524	3.517	28.2

**Table 5 sensors-20-07197-t005:** Results comparisons in the dynamic scenario.

Symbol	Description	HPFSM	PFBM	−(%)
$a_{y,\max}$ (m/s${}^{2}$)	Maximum lateral acceleration	0.293	2.410	87.8
$a_{y,\mathrm{mean}}$ (m/s${}^{2}$)	Average lateral acceleration	0.029	0.180	83.9
$\omega_{\max}$ (deg/s)	Maximum yaw rate	3.508	20.430	82.8
$\omega_{\mathrm{mean}}$ (deg/s)	Average yaw rate	0.477	1.713	72.2

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
