# Peer review of "Hybrid Path Planning Combining Potential Field with Sigmoid Curve for Autonomous Driving"

_sensors, 2020, doi:10.3390/s20247197_

Round 1

Reviewer 1 Report

The paper presents a potential field based approach for ackerman-like mobile vehicles. The authors presented the developed method in a comprehensive way, explaining the different components of the system, covering both static and dynamic environments. The desired path takes in consideration the smoothness and feasibility, while ensuring a collision free trajectory.

The reviewer finds that the paper faces an interesting challenge, but lacks to address major problems when dealing with similar scenarios. Firstly, instead of defining the trajectory, the shape of the potential field around the vehicles can be changed in an oval way so there won’t be any abrupt maneuvers while driving pass the vehicles. Secondly, the approach does not take in consideration both the “ego vehicle” speed and velocity, nor the ones of the vehicles. And finally, and most importantly, the authors claimed that the approach overcomes the problems of classical PF methods when it comes to local minima, but no explanation, nor simulation were conducted to support that claim.

Additional remark, in line 15, the classification of global and local path planning does not depend on the holonomic constraints of the vehicles, and the example of using PF for a non -holonmic vehicle is the best example, especially since originally PF were designed to control end effectors (robotic arms), which has 6 Degrees Of Freedom (DOF). The reviewer would also like to know which interpolation function the authors used in line 153 and what are its parameters.

Reviewer 2 Report

The article is devoted to an urgent and intensively discussed problem - the choice of a method for autonomous driving of a vehicle. The article provides a good overview of the main methods used for generating autonomous driving scenarios. The article proposes an original hybrid way of planning the path of an unmanned vehicle. This method is a combination of potential field and sigmoid curves. To confirm the effectiveness of the proposed method, the cases of movement of an unmanned vehicle with several obstacles are considered. The article is supplied with rich illustration material. There are comments on the article:

  1. Fig. 1 shows the equations, but there is no decoding of the symbols used.
  2. Row 92 indicates the coefficients a and b, but does not say what they are physically.
  3. fig. 3,4,5,8,9 are executed inaccurately - it is necessary to make arrows in the direction of the vehicle's trajectory.
  4. Table 2 uses different brackets. This makes it difficult to perceive the material.
  5. 5. The results in Table 5 are impressive! However, in the conclusions, it should be clarified that the results achieved were obtained on the basis of the analysis of the simulation results. With the actual use of the proposed route planning method, the figures are likely to be more modest.

The article is written on a relevant topic and at a sufficiently high level, contains all the necessary attributes, therefore, after eliminating the comments, the article can be published in "Sensors" journal.

Round 2

Reviewer 1 Report

In the first round, the reviewer required some clarifications concering the following points:

  • The PF shape
  • The vehicle (ego and obstacle) velocities when designing the PF
  • The local minima problem
  • The interpolation function

The authors successfully addressed the aforementioned points, thus the reviewer has no further remarks concerning the paper, and suggest its acceptance for publication in the journal of Sensors.